# Evaluating the predictive value of frailty scores for critical care admission and hospital stay in elderly surgical patients: A comparison of the mFI-5 and CCI

Marc Lincoln[1]*, Liadán Tobin-Schnittger[1], Marianne Foley[2], Dulmi Nawartha[1], Pádraig Ó Scanaill[1]

1 Department of Anaesthesiology, Mater Misericordiae University Hospital, Dublin, Ireland, 2 Department of Medicine for the Elderly, Mater Misericordiae University Hospital, Dublin, Ireland

* lincolnmarc@gmail.com

## Abstract

### Background

Frailty is a critical determinant of postoperative outcomes in elderly patients. Several frailty assessment tools, including the Modified Frailty Index (MFI-5) and the Charlson Comorbidity Index (CCI), have been proposed to predict complications, hospital length of stay (LoS), and critical care admission. However, their comparative predictive value across a broad spectrum of non-cardiac surgeries remains unclear. The purpose of this study was to assess the predictive ability of MFI-5 and CCI in predicting critical care admission and length of stay (LoS).

### Methods

This single-centre retrospective study analysed data from patients over 65 years of age who attended the preoperative assessment clinic at the Mater Misericordiae University Hospital (MMUH), Dublin, between November and December 2023. MFI-5 and CCI scores were calculated, and their ability to predict hospital LoS (>5 days) and critical care admission was assessed using area under the receiver operating characteristic curve (AUROC) analysis.

### Results

Data from 100 patients were included. Critical care admission was required for 20 patients, and the average hospital length of LoS was 4.5 days. AUROC analysis demonstrated that neither the MFI-5 nor CCI were predictive of critical care admission or extended LoS in this cohort.

**Data availability statement:** Data cannot be shared publicly because of GDPR restrictions. Data are available from the ethics committee at the MMUH +353 1 803 2971 hannahking@mater.ie for researchers who meet the criteria for access to confidential data.

**Funding:** The author(s) received no specific funding for this work.

**Competing interests:** No authors have competing interests.

## Conclusion

The findings suggest that MFI-5 and CCI alone may not be sufficient to predict critical care admission or hospital LoS in elderly patients undergoing non-cardiac surgery. Given the multifactorial nature of postoperative risk, future models integrating frailty indices with surgical and anaesthesia-specific factors may enhance predictive accuracy, improve risk stratification, and optimize perioperative resource allocation.

## Introduction

The aging process can lead to frailty, a state of reduced resilience across multiple body systems [1]. Given the growing number of older adults undergoing surgery, frailty is a significant concern. It has been clearly established that frailty independently increases the risk of complications after surgery [2–4].

Several frailty assessment tools have been developed to predict various patient outcomes. The Charlson Co-morbidity Index (CCI) was developed in 1987 and was initially used to predict mortality in frail patients [1]. More recently, a modified 11-item frailty index has been created using data from the Canadian study of Health and Aging frailty index (CSHA-FI) and is based on a cumulative deficit model [2,3]. From this, a more practical clinical tool based on five clinical characteristics has been developed known as the modified 5-item frailty index (mFI-5) and has been shown to be equally predictive for 30-day outcomes as the mFI-11 [4].

In surgical patients, there's growing interest in using risk scores to predict clinical outcomes beyond just mortality, including hospital length of stay, readmission rates, the need for intensive care, and post-operative complications [9–17]. Although studies have examined the predictive power of these scores for ICU admissions and hospital stays in specific surgical contexts, a direct comparison of two key scores across a broad spectrum of non-cardiac surgeries is lacking. Predicting which patients will require critical care or have extended hospital stays is crucial for hospitals to efficiently allocate limited resources, ensuring timely care for critical patients while controlling costs. Identifying at-risk patients early allows for personalized care plans, potentially leading to improved outcomes and lower readmission rates. Additionally, this foresight enables healthcare providers to set realistic expectations for patients and their families, assisting in their emotional and logistical preparation.

In this study, we compared the mFI-5 and CCI in their predictive value for hospital length of stay (LoS) and critical care admission.

## Methods

This was a single-centre retrospective study undertaken at the Mater Misericordiae University Hospital (MMUH), Dublin between November and December 2023. The clinical endpoints were admission to critical care at any point during hospital admission and length of stay (LoS) more than five days. Critical care encompassed either admission to an intensive care unit (level 3 care) or a high dependency unit (level 2 care). The research was approved by the research ethics committee (REC) at MMUH (Ref: 1/378/2445).

## Data collection

All data was collected retrospectively for patients over the age of 65 who attended the pre-operative assessment clinic in November and December 2023. All patients were referred prior to elective admission for surgery. The electronic patient record for each patient was accessed in order to acquire basic demographic information as well as the American Association of Anaesthesiologists (ASA) score. Data was collected on November 17, 2024. All data was anonymised and patient consent waived by the ethics committee in line with GDPR guidelines.

The CCI and MFI-5 were calculated for each patient. For example, a patient affected by diabetes mellitus, hypertension requiring medication and cerebrovascular accident with deficit was scored with a mFI-5 of 3. A patient aged 84 with a history of COPD would get a CCI score of 5.

## Statistical analysis

We compared each of the mFI-5 and CCI scores in their ability to predict critical care admission and LoS using calculation of area under the receiver operator curve (AUROC) analysis and 95% confidence intervals. Data analysis was undertaken using SPSS. We anticipated approximately 120 patients over 65 would attend the preoperative assessment clinic over two months, based on previous audits. A power analysis, using a 30% event rate and AUROC threshold of 0.7, indicated that a sample size of about 100 would be sufficient and this was in line with previous research in the area [5].

## Results

Data from 100 patients was collected. 62 were male and 38 were female. The average age was 74 and the range was 66−84. 20 patients required critical care during their admission and the average length of stay was 4.5 days with the maximum length being 70 days. Further demographic information is outlined in Table 1 along with CCI and mFI-5 score for all patients in Table 2.

AUROC analysis (Table 3) was performed to determine the predictive value of MFI-5 and CCI on LoS and admission to critical care. The AUROC for MFI-5 and CCI were 0.62 and 0.59 respectively for length of stay and 0.52 and 0.53 respectively for critical care admission.

## Discussion

Our study found that the Modified Frailty Index (MFI-5) and Charlson Comorbidity Index (CCI) were not predictive of critical care admission or hospital length of stay in elderly postoperative patients. Originally developed to assess mortality risk through comorbidity severity, these scores primarily serve as indirect indicators of physiological reserve. While reduced physiological reserve is known to increase postoperative morbidity, these indices may lack sensitivity to other key predictors of postoperative outcomes [6]. Specifically, factors such as the type and duration of surgery and the mode of anaesthesia significantly impact postoperative morbidity, influencing both critical care admission and hospital stay. Although our study included a heterogeneous mix of surgeries, these factors are not captured by the MFI-5 or CCI, potentially explaining their limited predictive value. Future tools that integrate frailty indices with surgical and anaesthesia-specific factors could more accurately predict critical care needs and recovery timelines in elderly patients, improving high-risk patient identification and enhancing perioperative planning and resource allocation.

Interestingly, these scores have shown some predictive ability in specific surgeries. For example, the MFI-5 has predicted length of stay in orthopaedic surgery including head and neck surgery and shoulder, elbow, hip and knee arthroplasty [7–10] but failed in gynaecologic oncology and lumbar decompression surgery [11,12]. It was able to predict ICU admission in patients undergoing gastric and pancreatic cancer surgery, head and neck cancer surgery and PCNL although in these studies the MFI-5 was used to split patients into 'frail' or 'not frail' based on a high or low score rather than AUROC analysis to predictive continuous predictability of the score [13–16]. CCI also has mixed results. It was predictive of increased length of stay and critical care in a cohort of patients undergoing colorectal cancer surgery but

**Table 1. Demographic and clinical characteristics of the 100 patients included in the study.**

| | |
|---|---|
| **Total no. of patients** | 100 |
| **Mean age, yrs (range)** | 74 (66-84) |
| **Sex** | |
| Male | 62 |
| Female | 38 |
| **Length of stay** | Days |
| <5d | 68 |
| >5d | 32 |
| **Specialty** | |
| Vascular | 11 |
| Orthopoedics | 2 |
| Otolaryngology | 5 |
| Urology | 16 |
| General Surgery | 48 |
| Gynaecology | 4 |
| Other | 14 |
| **Co-morbidities** | |
| Diabetes | 17 |
| COPD | 30 |
| Cardiac Failure | 10 |
| Myocardial Infarction | 12 |
| Angina | 17 |
| Hypertension | 76 |
| Peripheral Vascular disease | 10 |
| Impaired cognition | 7 |
| Cerebrovascular accident | 10 |
| Neurological deficit | 3 |
| Connective tissue disorder | 2 |
| Peptic ulcer disease | 25 |
| Chronic kidney disease | 15 |
| Cancer | 49 |
| **ASA** | |
| 1 | 0 |
| 2 | 36 |
| 3 | 62 |
| 4 | 2 |

not in those undergoing hip replacement [17–20]. Therefore, while our study demonstrates that MFI-5 and CCI were not predictive of length of stay and critical care admission in a wide range of surgeries, there are specific types of surgeries where it has predictive value. This discrepancy may be due to the heterogeneity of our study population, compared to the more homogeneous cohorts in other studies, as well as the lower physiological demands of the procedures in our cohort compared to more complex surgeries. In elective orthopaedic surgery, for example, frailty may have a more direct impact on recovery and length of stay, as other variables like surgical complexity and anaesthesia risks are minimized. Unlike prior investigations focused on individual surgical specialties, our study evaluated frailty indices in a general perioperative population. This broader approach is intended to mirror the pragmatic conditions of preoperative clinics,

**Table 2. CCI and mFI scores for patients attending POAC.**

| CCI | |
|---|---|
| Score | No. of patients |
| 0-5 | 44 |
| 6-10 | 51 |
| 11-15 | 5 |
| **mFI-5** | |
| Score | No. of patients |
| 0-1 | 50 |
| 2-3 | 39 |
| 3-4 | 11 |

**Table 3. AUROC analysis for critical care admission and length of stay.**

| Critical Care Admission | | |
|---|---|---|
| Score | AUROC (95%CI) | P value |
| CCI | 0.53 (0.39–0.78) | 0.66 |
| MFI | 0.52 (0.39–0.79) | 0.87 |
| **Length of Stay** | | |
| Score | AUROC (95%CI) | P value |
| CCI | 0.59 (0.43-0.72) | 0.22 |
| MFI | 0.62 (0.44–0.76) | 0.14 |

where surgical diversity and variable physiological demands often preclude the use of procedure-specific prediction models. As such, our findings provide insight into the real-world limitations of these indices when applied to heterogeneous patient cohorts.

Another potential explanation for the lack of predictive significance in frailty scores for critical care admission in this cohort may stem from the elective nature of the surgeries. In elective cases, clinicians have the opportunity to carefully assess patients preoperatively, and this may lead to a more conservative approach to postoperative care for patients identified as frail or at higher risk. Consequently, some patients may have been admitted to the critical care postoperatively as a precaution rather than out of immediate necessity. This pre-emptive decision-making likely reflects clinicians' efforts to mitigate potential complications rather than a response to acute postoperative deterioration. As a result, ICU admission in this context may not accurately represent a direct correlation with frailty or postoperative risk but instead an institutional preference or precautionary measure. This selection bias could have diluted the ability of frailty scores to accurately predict true ICU needs in this population.

Our sample size was limited by the two-month data collection period, chosen to reflect a practical and feasible window for capturing preoperative clinic activity. While this provided useful preliminary insights, a larger sample size may have yielded more precise estimates of predictive performance and allowed for subgroup analyses.

## Conclusion

The limited predictive value of these widely used frailty scores in this study demonstrates that relying solely on frailty indices may lead to underestimating or overestimating resource needs, which can impact staffing, ICU bed availability, and postoperative care planning. Instead, a more comprehensive approach that includes more individualized risk assessments, and consideration of surgery-specific factors (such as surgical complexity, expected blood loss, and anaesthesia type) may be required to better anticipate critical care needs and optimize recovery timelines. This study suggests that a

shift towards personalized perioperative care could improve patient outcomes and streamline resource allocation, helping to better meet the unique demands of elderly surgical patients.

## Author contributions

**Conceptualization:** Marc Lincoln, Marianne Foley, Pádraig Ó Scanaill.

**Data curation:** Marc Lincoln, Liadan Tobin-Schnittger, Marianne Foley.

**Formal analysis:** Marc Lincoln.

**Investigation:** Marc Lincoln.

**Methodology:** Marc Lincoln, Pádraig Ó Scanaill.

**Project administration:** Marc Lincoln, Dulmi Nawartha, Pádraig Ó Scanaill.

**Resources:** Marc Lincoln.

**Supervision:** Pádraig Ó Scanaill.

**Validation:** Pádraig Ó Scanaill.

**Writing – original draft:** Marc Lincoln.

**Writing – review & editing:** Marc Lincoln, Pádraig Ó Scanaill.

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
