## [Decision Letter · Decision Letter 0]

Jun 21 2025

PONE-D-25-15888Evaluating the predictive value of frailty scores for critical care admission and hospital stay in elderly surgical patients: a comparison of the mFI-5 and CCIPLOS ONE

Dear Dr. LINCOLN,

Thank you for submitting your manuscript to PLOS ONE. After careful consideration, we feel that it has merit but does not fully meet PLOS ONE’s publication criteria as it currently stands. Therefore, we invite you to submit a revised version of the manuscript that addresses the points raised during the review process.

We look forward to receiving your revised manuscript.

Kind regards,

Pedro Kallas Curiati, M.D., Ph.D.

Academic Editor

PLOS ONE

Journal Requirements:

3. Your abstract cannot contain citations. Please only include citations in the body text of the manuscript, and ensure that they remain in ascending numerical order on first mention.

4. Please remove all personal information, ensure that the data shared are in accordance with participant consent, and re-upload a fully anonymized data set.

Reviewers' comments:

Reviewer's Responses to Questions

**Comments to the Author**

1. Is the manuscript technically sound, and do the data support the conclusions?

Reviewer #1: Partly

Reviewer #2: Yes

Reviewer #3: Yes

2. Has the statistical analysis been performed appropriately and rigorously? 

Reviewer #1: Yes

Reviewer #2: Yes

Reviewer #3: Yes

3. Have the authors made all data underlying the findings in their manuscript fully available?

Reviewer #1: No

Reviewer #2: Yes

Reviewer #3: Yes

4. Is the manuscript presented in an intelligible fashion and written in standard English?

Reviewer #1: Yes

Reviewer #2: Yes

Reviewer #3: Yes

5. Review Comments to the Author

Reviewer #1: This is a reasonable piece of work which could be improved with revisions.

The conclusions are an accurate reflection of the result, showing that MFI-5 and CCI do not predict critical care or length of stay. However, it would be fair to say that neither of these tools were designed specifically for that purpose.

Recommend a better explanation of the sample size. Why only 2 months’ patients when presumably more data was available by November 2024? It is possible that with a large sample, clearer results would have been obtained. Were ALL patients who attended the clinic over 65 included, or was there a patient opt-out or other exclusion criteria?

Why was 5 days chosen as the cut-off for length of stay? Were other lengths considered and analysed?

I infer that all the patients were elective admissions to hospital, but this could be made explicit.

Not clear why HFRS (Gilbert https://doi.org/10.1016/S0140-6736(18)30668-8) was not mentioned. Also a recent paper (Kutrani https://doi.org/10.1371/journal.pone.0317234) showed that HFRS (or HFRS in conjunction with CCI) is effective at predicting LOS.

The paper is not explicit on what statistical or machine learning technique was used for the predictive model. Was it logistic regression? Also, how were the confidence intervals and p-values obtained?

The results show the demographics of the patients but not how many scored what mFI-5 or CCI scores (perhaps present this in suitable bands).

The results show little to no predictive ability of the models and the discussion does make a good attempt at explaining why this might be the case. The sentence, "This discrepancy may be due to the homogeneity of the patient population and the lower physiological demands of the procedure compared to more complex surgeries." might be clearer if it compared the heterogeneity of THIS study with the homogeneity of other studies.

In the discussion, other papers that the authors might wish to look at include:

• Goshtasbi (https://doi.org/10.1177/01945998211010443) found that the mFI-5 was associated with longer length of stay, medical complications, and mortality.

• Strigenz (https://doi.org/10.22603/ssrr.2022-0102) concluded that the mFI-5 was an independent predictor of morbidity, suggesting an increased risk of complication events (increased complications means longer LOS and more risk of admission to ICU).

Reviewer #2: Overall:

The study addresses an important topic, but the current manuscript has several limitations that limit its impact and interpretability. The negative findings are not surprising given the small sample size and heterogeneous surgical population, and the analysis lacks depth.

Major Concerns:

• The sample size (n=100) is underpowered to detect meaningful differences, particularly with only 20 ICU admissions.

• The analysis relies solely on AUROC without exploring more clinically meaningful cutoffs (e.g., frail vs non-frail), limiting applicability.

• The heterogeneity of surgical procedures likely masks any predictive signal; no subgroup or sensitivity analyses are provided.

• The manuscript does not offer sufficient novel insight beyond prior studies that have shown mixed results for mFI-5 and CCI in specific surgeries.

Reviewer #3: This is an automated report for PONE-D-25-15888. This report was solicited by the PLOS One editorial team and provided by ScreenIT.

ScreenIT is an independent group of scientists developing automated tools that analyze academic papers. A set of automated tools screened your submitted manuscript and provided the report below. Each tool was created by your academic colleagues with the goal of helping authors. The tools look for factors that are important for transparency, rigor and reproducibility, and we hope that the report might help you to improve reporting in your manuscript. Within the report you will find links to more information about the items that the tools check. These links include helpful papers, websites, or videos that explain why the item is important. While our screening tools aim to improve and maintain quality standards they may, on occasion, miss nuances specific to your study type or flag something incorrectly. Each tool has limitations that are described on the ScreenIT website. The tools screen the main file for the paper; they are not able to screen supplements stored in separate files. Please note that the Academic Editor had access to these comments while making a decision on your manuscript. The Academic Editor may ask that issues flagged in this report be addressed. If you would like to provide feedback on the ScreenIT tool, please email the team at ScreenIt@bih-charite.de. If you have questions or concerns about the review process, please contact the PLOS One office at plosone@plos.org.

6. PLOS authors have the option to publish the peer review history of their article (what does this mean? ). If published, this will include your full peer review and any attached files.

**Do you want your identity to be public for this peer review?** For information about this choice, including consent withdrawal, please see our Privacy Policy .

Reviewer #1: No

Reviewer #2: No

Reviewer #3: No

---

## [Author Response · Author response to Decision Letter 1]

12 May 2025

Reviewer #1:

Many thanks for taking the time to review this paper. I hope you will find the ammendments and responses satisfactory. Marc

•

The conclusions are an accurate reflection of the result, showing that MFI-5 and CCI do not predict critical care or length of stay. However, it would be fair to say that neither of these tools were designed specifically for that purpose.

Response: This is a fair point and I have included a sentence in the discussion which now addresses this.

•

Recommend a better explanation of the sample size. Why only 2 months’ patients when presumably more data was available by November 2024? It is possible that with a large sample, clearer results would have been obtained.

Response: We limited data collection to a two-month period (November–December 2023) to reflect a practical snapshot of patient flow through the preoperative assessment clinic, based on available resources and feasibility. A retrospective power analysis using an anticipated event rate of 30% and AUROC threshold of 0.7 suggested that a sample size of approximately 100 patients would provide sufficient power to detect moderate predictive ability, consistent with similar published studies in this field as referenced in the article. However, we acknowledge that this sample size is modest, and the wide confidence intervals observed in our AUROC results likely reflect this. It is possible that with a larger sample size, clearer patterns may have emerged regarding the predictive value of the frailty indices. Future prospective studies with larger cohorts could offer more definitive conclusions and allow for subgroup analysis by surgical specialty or urgency. I have added a short paragraph into the discussion section which acknowledge this limitation to our study.

• Were ALL patients who attended the clinic over 65 included, or was there a patient opt-out or other exclusion criteria?

Response: Yes every patient over the age of 65 was included and there were no exclusion criteria. I have exmphaised in the amended manuscript for clarity.

•

Why was 5 days chosen as the cut-off for length of stay? Were other lengths considered and analysed?

Response: We selected a length of stay cut-off of 5 days prior to conducting the study, based on local clinical expectations and its use as a marker of prolonged admission in similar surgical cohorts. We did not test multiple thresholds post hoc, due to concern over false positive results.

•

I infer that all the patients were elective admissions to hospital, but this could be made explicit.

Response: I have included a sentence on this in the methods section for clarity.

•

Not clear why HFRS (Gilbert https://doi.org/10.1016/S0140-6736(18)30668-8) was not mentioned. Also a recent paper (Kutrani https://doi.org/10.1371/journal.pone.0317234) showed that HFRS (or HFRS in conjunction with CCI) is effective at predicting LOS.

Response: I agree that the Hospital Frailty Risk Score (HFRS) is an important tool in the assessment of frailty, particularly in large administrative datasets. However, our study was based on clinical data collection rather than ICD coding, which the HFRS relies upon, and thus we didn’t feel it was feasible to implement in our setting. Furthermore, we felt that CCI and MFI-5 are more useful in the point of care surgical setting like the per-operative care setting in which are study takes place rather than HFRS. I haven’t included reference to this in the amended manuscript as it stands but happy to reference it if you feel it would improve the validity of the paper.

•

The paper is not explicit on what statistical or machine learning technique was used for the predictive model. Was it logistic regression? Also, how were the confidence intervals and p-values obtained?

Response: To assess the predictive ability of the frailty scores we performed AUROC analysis. This method was used to evaluate the discriminatory ability of each score in predicting outcomes. No logistic regression or machine learning models were used in this analysis. SPSS was used for all statistical analysis and I did this myself. The 95% confidence intervals and p values for AUROC on SPSS is calculated based off the Hanley and Mcneil mtehod. P-values were derived from the comparison of AUROC values with the null hypothesis that the score has no discriminative ability (i.e., AUROC = 0.5). A p-value of less than 0.05 was considered statistically significant, although none of the tests reached significance.

I feel I have included enough information in the methods section on the statistical analysis to be in line with other papers on similar topics but I am more than happy to go into more detail in the manuscript if you feel it is necessary for clarity.

•

The results show the demographics of the patients but not how many scored what mFI-5 or CCI scores (perhaps present this in suitable bands).

Response: I have added in a table into the methods section for this.

•

The results show little to no predictive ability of the models and the discussion does make a good attempt at explaining why this might be the case. The sentence, "This discrepancy may be due to the homogeneity of the patient population and the lower physiological demands of the procedure compared to more complex surgeries." might be clearer if it compared the heterogeneity of THIS study with the homogeneity of other studies.

Response: Many thanks for this comment. I agree it is not clear, and I have amended it in the discussion to draw a more clear comparison between the heterogeneity of this study population and the homogeneity of other studies.

•

In the discussion, other papers that the authors might wish to look at include:

• Goshtasbi (https://doi.org/10.1177/01945998211010443) found that the mFI-5 was associated with longer length of stay, medical complications, and mortality.

• Strigenz (https://doi.org/10.22603/ssrr.2022-0102) concluded that the mFI-5 was an independent predictor of morbidity, suggesting an increased risk of complication events (increased complications means longer LOS and more risk of admission to ICU).

Response: I have included reference in the discussion to the paper by Goshtasbi. I agree that Stigenz’s paper conluded that MFI-5 was a predictor of morbidity but given it didn’t specifically look at ICU admission or length of stay I didn’t include it in the reference list if that’s okay.

Reviewer #2:

Many thanks for taking the time to review this paper, I hope you will find the responses and subsequent ammendments to the manuscript acceptable. Marc

• The sample size (n=100) is underpowered to detect meaningful differences, particularly with only 20 ICU admissions.

Repsonse: Thank you for this important comment. A power analysis was conducted prior to study initiation to ensure adequate planning; however, we acknowledge that the final sample size remains modest, particularly in relation to the relatively small number of ICU admissions. We have now addressed this limitation explicitly in the discussion section, and we note that larger sample sizes in future studies may improve the stability and generalizability of predictive estimates.

• The analysis relies solely on AUROC without exploring more clinically meaningful cutoffs (e.g., frail vs non-frail), limiting applicability.

Response: Thank you for this valuable observation. We chose AUROC as our primary analytic tool because it offers a comprehensive measure of a model’s overall discriminative ability across all possible thresholds. While we recognize that dichotomous classifications (e.g., frail vs. non-frail) may have clinical relevance, our intent was to assess the overall predictive value of these indices which can be generalised across diverse surgical populations.

• The heterogeneity of surgical procedures likely masks any predictive signal; no subgroup or sensitivity analyses are provided.

Response: We appreciate this comment and agree that procedural heterogeneity may have reduced the predictive signal of frailty indices and the modest sample size constrained our ability to conduct formal subgroup analyses with sufficient power. We have now addressed this in the discussion section pointing out how the heterogeneity of our sample population is an explanation for the lack of predictive value of the scoring systems in this paper compared with other papers which show predictive value of them in much more homogenous populations, such as specific surgeries.

• The manuscript does not offer sufficient novel insight beyond prior studies that have shown mixed results for mFI-5 and CCI in specific surgeries.

Response: Thank you for this feedback. While we acknowledge prior studies have examined mFI-5 and CCI in specific surgical contexts, our study aimed to assess the feasibility and predictive utility of these indices in a general perioperative clinic setting, where patient populations are diverse and decisions about preoperative optimization must often be made in the absence of procedure-specific data. This broader approach adds practical insight into how frailty tools perform in real-world, mixed-cohort settings. We have revised the discussion adding a short paragraph which better articulate this distinction and to clarify how our findings contribute to the ongoing evaluation of frailty measures in everyday perioperative practice.

Reviewer #3

ScreenIT has been reviewed, and suitable ammendments to manuscript have been made.

---

## [Editor Report · Decision Letter 1]

Evaluating the predictive value of frailty scores for critical care admission and hospital stay in elderly surgical patients: a comparison of the mFI-5 and CCI

PONE-D-25-15888R1

Dear Dr. LINCOLN,

We’re pleased to inform you that your manuscript has been judged scientifically suitable for publication and will be formally accepted for publication once it meets all outstanding technical requirements.

Kind regards,

Pedro Kallas Curiati, M.D., Ph.D.

Academic Editor

PLOS ONE
---

## [Editor Report · Acceptance letter]

PONE-D-25-15888R1

PLOS ONE

Dear Dr. LINCOLN,

I'm pleased to inform you that your manuscript has been deemed suitable for publication in PLOS ONE. Congratulations! Your manuscript is now being handed over to our production team.

Kind regards,

on behalf of

Dr. Pedro Kallas Curiati

Academic Editor

PLOS ONE